# Microclimate of Grape Bunch and Sunburn of White Grape Berries: Effect on Wine Quality

**DOI:** 10.3390/foods12030621

**Published:** 2023-02-01

**Authors:** Laura Rustioni, Alessio Altomare, Gvantsa Shanshiashvili, Fabio Greco, Riccardo Buccolieri, Ileana Blanco, Gabriele Cola, Daniela Fracassetti

**Affiliations:** 1Department of Biological and Environmental Sciences and Technologies, University of Salento, Via Provinciale Monteroni, 73100 Lecce, Italy; 2Department of Food, Environmental and Nutritional Sciences (DeFENS), Università degli Studi di Milano, Via G. Celoria 2, 20133 Milan, Italy; 3Department of Agricultural and Environmental Sciences (DISAA), Università degli Studi di Milano, Via G. Celoria 2, 20133 Milan, Italy

**Keywords:** vineyard management, leaf removal, winemaking, phenols, antioxidant capacity, solar radiation, climate change, orange/amber wines

## Abstract

This research aimed to evaluate the composition of wines made with white grapes which are particularly susceptible to sunburn symptoms due to the absence of anthocyanin. Sunburn is a complex physiological dysfunction leading to browning or necrosis of berry tissues. In vintage 2021, the canopy of ‘Verdeca’ grapevines grown in Salento, South Italy, was differently managed by sun exposing or shading the bunches. Micrometeorological conditions were studied at different levels. Grapes were vinified, comparing the winemaking with and without skin maceration. The vegetative-productive balance of plants was not substantially modified. On the contrary, a significant effect was observed on the quality and quantity of grapes produced: smaller berries with sunburn symptoms were found on unshaded bunches. This influenced the percentage distribution among skin, pulp and seeds, causing a decrease in must yield of up to 30%. The pH was significantly higher in macerated wines made using shaded grapes, due to a lower titratable acidity and to significant impacts on the acid profile. Obviously, maceration produced a higher extraction of phenolics in wines, which reached their maximum in wines made with sunburned grapes. The absorbance at 420 nm, index of yellow color, was also significantly higher in sunburned grapes, indicating greater oxidation. Even though excessive grape sun-exposure could negatively affect the perception of white wines made without maceration (resulting in more oxidative character), the sensory quality of orange/amber wines was not significantly impacted by the presence of sunburned grapes. Thus, this winemaking technique could be particularly interesting to set up a production strategy adapted to viticultural regions strongly affected by climate change.

## 1. Introduction

Vineyard management significantly affects grape quality. The effects of irrigation techniques, soil management, winter pruning and canopy management have been widely investigated [1,2,3,4]. Considering leaf removal, different studies have shown significant effects of the bunch exposure to sunlight on the grape quality, mainly recording an increase in secondary metabolites, such as phenolics [5,6,7]. It is well known that these molecules play an important role in the protection against radiative excesses [8,9,10,11,12]. In fact, sunlight excess, often associated with high temperatures, could cause different kinds of physiological disorders, commonly recognized as sunburn damages. Incidence and severity of the symptom depends on a complex interplay of environmental factors and on the various response mechanisms of the plant tissues to the stress [13]. Generally, during sunburn, photosynthetic pigments undergo degradation [14]. Light and, in general, stresses induce the accumulation of secondary metabolites, such as phenolics [15], and the oxidative stress associated with a cell decompartmentalization causes phenolic oxidation. This leads to the accumulation of brown pigments [12,16]. Thus, changes in the metabolisms induce differences in the chemical composition of the grapes, with impacts on the quality of production [17]. In sunburned grapes, the crystalline structure of the waxes undergoes degradation to amorphous masses, affecting the protective role of this layer [18]. As a consequence, the loss in wax protection properties and the tissue necrosis (with consequent cracking) favor berry desiccation [13]. In extreme cases, shriveling of entire berries, and even of entire bunches, can occur, also affecting parts of the rachis [13].

Both viticultural and oenological decisions are interconnected in the wine industry, influencing each other. Thus, the optimization of vineyard management according to the oenological objective, as well as the valorization of the harvested grapes with the most suitable winemaking techniques, are of paramount importance. Relatively few studies take into account the effects of vineyard canopy management on food processing technologies. For example, Rustioni et al. [19] showed a delay in anthocyanin extractability related to bunch exposure. This can be due to a discrepancy between the technological maturity and the phenolic maturity. Piombino et al. [20] highlighted a significant effect of defoliation on the dynamics of dehydration of Nebbiolo grapes in post-harvest processing. In the case of sunburn of white berries, the presence of oxidized phenols in grapes could favor the oxidation of flavonols in wine [21], which can cause qualitatively deleterious browning or even pinking [22]. These aspects should be considered in the production of white wine both with and without skin maceration. Even if lower extraction of phenols is expected when no skin contact is performed, the presence of oxidized phenols in grapes could lead to a must richer in oxidized phenols also causing a possible faster decrease in antioxidants (i.e., reduced glutathione) [23]. As a consequence, the shelf life of these wines can be shortened. The production of white wine with skin maceration leads to higher content of extracted phenols, significantly contributing to stability of white wine, and a longer aging potential due to the major antioxidant capacity [24,25,26]. The skin maceration also affects the sensory characteristics as the perception of herbaceous notes can be more evident depending to the higher content of C6 volatile compounds [27]. The prolonged contact between grape skins and must-wine also affects the acidity as ions, such as potassium, which can be present at higher concentrations, are also responsible for the pH increase [24,27]. Nonetheless, the suitability of white grape varieties towards skin maceration should be also considered with the purpose of implementing the market of new style wines having the desired characteristics. In this scenario, when grape berries with sunburn symptoms are processed following a hot and sunny season as occurred in vintage 2022 in Italy, the extraction of oxidized phenols can have a considerable impact as well.

To the best of our knowledge, the effects of sunburn symptoms on wine quality have not yet been investigated. Thus, further research is needed in order to clarify this aspect. The aim of this study was to provide evidence for the interactions between viticulture and oenological decisions, comparing two canopy managements performed in an extremely stressing growing area (South Italy)—inducing bunch exposure and bunch shading, respectively—and two winemaking techniques—with and without skin and seed maceration. The impact of notable sunlight exposure was evaluated in grape bunches in relation to the microclimate. Experimental wines were produced, with and without skin maceration, in order to evaluate the impact of sunburned grapes in an oenological perspective considering two winemaking procedures.

## 2. Materials and Methods

### 2.1. Experimental Plan

Figure 1 shows a summary of the experimental plan in a flow chart.

The experiment was conducted in a vineyard belonging to Azienda Vitivinicola Marulli-Copertino, LE, Southern Italy (latitude: 40°18′01″ N; longitude: 18°02′19″ E; altitude: 40 m a.s.l.). In the Köppen–Geiger classification, Copertino belongs to the Hot-summer Mediterranean Climate Csa class, with dry and hot summers, due to the dominance of subtropical high-pressure systems, and mild and wet winters with moderate and changeable temperatures. The Mediterranean region is potentially vulnerable to climatic changes because it is affected by interactions between mid-latitude and tropical processes [28]. The experimental site, in the middle of the Mediterranean Sea, is considered a particular hot spot for climate change impacts, being already strongly stressful for grapevine cultivation. The cultivar grown is ‘Verdeca’, an Apulian local white grape variety. The vineyard is planted in a plain area and row orientation is north/south. Plants are spaced 2.0 m (interrow) and 1.1 m (intrarow), with a plant density of about 4500 plants/ha. Vines are pruned using the classic Guyot system. Soil is managed by tillage, and the vineyard is not irrigated.

The experiment involved 6 rows of about 130 m each, and it consisted of:4 June 2021: Vine topping at fruit set to induce thickening of the leaf wall by growth of secondary shoots.17 June 2021: Vertical positioning of the shoots and installation of the micrometeorological sensors.13 July 2021: At bunch closure, leaf (and secondary shoot) removal in the bunch zone to ensure a bunch exposure of 100% and to induce sunburn symptoms. This treatment was performed only on 3 alternative rows, while the others were left without defoliation, with a bunch exposure of about 10% (examples of the canopy obtained with the two treatments are available in Appendix A).27 July 2021: At the end of the veraison, canopy description.11 September 2021: Grape harvesting for winemaking and for carpological analysis and description of sunburn symptoms.

### 2.2. Canopy Description

In each of the 6 rows, 5 groups of 5 plants were observed. In details, measurements in each group of plants included:leaf wall height;number of leaf layers (5 replications);estimation of the percentage of empty spaces;number of shoots in active growth;number of primary shoots in 1 plant;number of bunches in 1 plant;number of primary shoot leaves in 1 shoot;number of secondary shoots and their leaves in 1 primary shoot.

The leaf areas of 15 primary shoot leaves and 15 secondary shoot leaves were measured singularly by using the smartphone app “Easy Leaf Area Free” [29], as described by Dinu et al. [30]. These data were used to estimate the total leaf area based on the number of leaves of each plant. Furthermore, the percentages of the leaf area due to primary shoot leaves and secondary shoot leaves were estimated. Finally, the vegeto-productive balance was quantified as leaf area/bunch.

### 2.3. Sensors and Micrometeorological Data Analyses

Air temperature and relative humidity measurements were carried out continuously in the period 13 July 2021–11 September 2021 using Gemini Data Loggers Tinytag Plus 2-TGP-4505, which are rugged, waterproof data loggers with temperature and relative humidity probes with an accuracy of ±0.3 °C (air temperature) and ±3% (relative humidity) at 25 °C. Specifically, one probe was positioned at about 2 m a.g.l. to measure the ambient temperature and relative humidity, while two other probes were placed inside the grape bunches (in one of the rows with defoliation and in one of the rows without defoliation, respectively) at the early stage of ripening, and care was taken not to break the berries. This was considered a reasonable compromise to measure the temperature and humidity inside the grape bunch to which it is exposed. The sensors stored 15 min average values.

In order to represent the outer temperature of the berries, a simulation of the temperature of the berries was further performed by means of the modeling tool BerryTone described by Cola et al. [31]. BerryTone uses a mechanistic approach founded on the energy balance, and it is driven by maximum and minimum daily air temperatures. The model considers the effect of different sun exposures and leaf shadings.

### 2.4. Carpological Analysis and Description of Sunburn Symptoms

At harvesting time, a sample of about 750 g of berries was collected in each of the 6 rows (3 replications/treatment). The berries of each sample were classified based on the sunburn symptoms as:asymptomatic berries;amber-colored berries;severely damaged berries;completely dry berries.

Photos of this classification are available in the Appendix A.

For each sample, the berries belonging to each group were counted and weighed. Furthermore, for each group of berries, 10 representative berries were used for carpological analysis, measuring the weights of the entire berries, of the skins and of the seeds. The number of seeds was also counted. These data were used to estimate the sunburn damages, as well as the impact of the treatments on the proportions among liquid (pulp) and solid (dry berries, skins and seeds) fractions.

### 2.5. Winemaking Process

Verdeca grapes were manually harvested by using 10 kg boxes. In each experimental row, 10 kg of grapes were randomly collected, resulting in 30 kg of both non-sunburned and sunburned grapes in total. Grapes were kept separated for the winemaking process that followed the same procedure. The pressing was carried out on the whole bunches using a vertical hydropress with lateral membrane having a capacity of about 80 kg (Hydro 80, Zambelli Enotech, Camisano Vicentino, VI, Italy). The pressing was performed at 3 bar for about 30 min; then, the grapes were manually mixed up and a second pressing step was carried out at 3 bar for about 30 min. At the end of pressing, the grape pomace was collected and stored at 4 ± 1 °C. The musts were stored at 4 ± 1 °C overnight for static settling. The content of readily assimilable nitrogen (RAN) was determined and diammonium phosphate (DAP) was added up to 200 mg/L. Each must was split in 6 aliquots, of which 3 aliquots were vinified without grape pomace (4 L each). The remaining 3 aliquots were added with 30% (*w*/*v*) of grape pomace (2.5 L of must added with 750 g of grape pomace). Commercial dry *Saccharomyces cerevisiae* EC1118 yeast strain was inoculated (10^6^ CFU/mL) in all the trials following the procedure suggested by the producers. Briefly, 6.6 g of dry yeast was re-hydrated with 500 mL of water and left under stirring for about 20 min; next, 1 L of clear Verdeca must was added, and it was left for about 20 min at room temperature. In each bottle containing 4 L of must, 200 mL of inoculum solution was placed. The alcoholic fermentation was carried out at 22 ± 1 °C, and it was monitored daily by weight loss; once about 30% of sugars were consumed, an additional addition of DAP (50 mg/L) was carried out. The alcoholic fermentation was considered completed when no weight change was observed after two consecutive days that was evidenced by the analysis of residual sugars. At the end of fermentation, the bottles were stored at 4 ± 1 °C for settling. The wines were racked, bottled and 100 mg/L of potassium metabisulfite was added. At racking, the skins of macerated wines were manually pressed in order to recover the wine. A total of 12 wines were produced that were stored at 4 ± 1 °C for 4 months before the chemical and sensory analysis.

### 2.6. Chemical Analysis

Glucose, fructose, ammonium and α-amino nitrogen, the sum of which corresponds to RAN, were determined using an automatic enzymatic analyzer iCUBIO i-Magic M9 (R-Biopharm, Melegnano, Italy) with specific assay kits for d-fructose/d-glucose, ammonium and α-amino nitrogen according to the manufacturer’s instructions. For the abovementioned kits, the accuracy, expressed as relative standard deviation (RDS%), is equal to 1.3% for the d-fructose/d-glucose and ammonium kits, and 1.5% for the α-amino nitrogen kit.

The total acidity was determined by titration up to pH 7 in accordance with the method OIV-MA-AS313-01 [32].

The quantification of ethanol was carried out by Enoconsulting (Erbusco, BS, Italy), an ISO 9000-accredited laboratory, through density [33].

The total phenol index (TPI) was determined following the Folin–Ciocalteu method [34,35]. Briefly, the wines were diluted up to 5-fold and up to 10-fold for the wines obtained without maceration and with maceration, respectively, in methanol/water 50/50 (*v*/*v*). The Folin–Ciocalteu reagent was diluted 10 times in water (*v*/*v*) and 2.5 mL was added to 0.5 mL of sample. Two milliliters of 75 g/L sodium carbonate solution was added and the tubes were kept for 1 h at room temperature in the dark. In the meantime, the calibration curve for gallic acid (5–100 mg/L) dissolved in methanol/water 50/50 (*v*/*v*) was achieved. The absorbance at 765 nm was measured, and the results were expressed as mg gallic acid/L.

The total flavonoids were quantified for the samples of wine produced with maceration. The wine samples were diluted up to 4 times with hydrochloric ethanol solution in order to obtain an absorption value approaching lower than 1 AU at 280 nm. The spectra were recorded in the wavelength range 700–230 nm and the quantification was carried out according to Corona et al. [36] and Fracassetti et al. [35]. The results were expressed as mg catechin/L, considering the height of the peak registered at 280 nm. The spectrophotometric readings at 420 nm were carried out as a marker of yellow color.

The antioxidant capacity was measured by means of the DPPH assay following the method of Brand-Williams et al. [37], with some modifications as reported by Fracassetti et al. [35] and Piva et al. [38]. Briefly, the DPPH solution was diluted in methanol to obtain 1.00 ± 0.03 absorbance units at 515 nm. The wine samples were diluted in 70% methanol (*v*/*v*) up to 4-fold. The DPPH solution (2.94 mL) was placed in a cuvette where 60 µL of sample was added. The absorbance readings were carried out after incubation for 50 min at 20 ± 1 °C. A calibration curve was prepared by adding increasing concentration of Trolox ranging from 0.15 to 1.5 mmol/L; each concentration was assayed in triplicate, as well as each sample. Results were expressed as mmol Trolox/L.

The organic acids were quantified as described by Fracassetti et al. [39] with some modifications. An Acquity HClass UPLC (Waters, Milford, MA, USA) system equipped with a photo diode array detector 2996 (Waters) was used. Chromatographic separations were performed with a Synergy, 4 μm HYDRO-RP, 80 A, 250 × 4.6 mm (Phenomenex, Torrance, CA, USA). The separation was carried out in isocratic conditions using phosphate buffer 20 mM at pH 2.9 at a flow rate of 0.6 mL/min, and the column temperature was at 30 °C. Calibration curves were obtained for tartaric, malic, lactic, citric, acetic and succinic acids at concentrations of 0.1–10 g/L, giving a linear response in the concentration range considered. The detection limit (LOD) and the quantification limit (LOQ) of the method were 0.003 g/L and 0.010 g/L for LOD and LOQ, respectively. RSD% ranged from 3.4% for tartaric acid to 8.6% for succinic acid. The wine samples were filtered with PVDF 0.22 μm filter (Millipore, Billerica, MA, USA) prior to the injection. The detection was carried out at 210 nm. Chromatographic data acquisition and processing were performed by Empower 2 software (Waters).

### 2.7. Sensory Analysis

To demonstrate the possible sensory differences due to the sunburn, two triangle tests were carried out considering the wines produced with and without maceration. The panel was composed of 20 judges (average age 28; 11 females, 9 males) who were asked to indicate the different wine samples firstly considering only the taste and thereafter both color and taste. The indication of the reason why the sample was chosen as different was also reported. The samples were presented to the panelists in a coded randomized order under ambient temperature and light. Dark and clear ISO glasses were used containing 25 mL of samples, and they were covered with a plate.

### 2.8. Statistical Analysis

The statistical analysis was performed with SPSS Win 12.0 program (SPSS Inc., Chicago, IL, USA). One-way ANOVA was carried out for the assessment of significant differences related to the grape berries. Factorial ANOVA was carried out to determine the significant differences related to fermentation trend and the chemical parameters of experimental wines. Firstly, the multivariate test of significance was determined; secondly, the post-hoc Fischer LSD (α = 0.05) was carried out. For the triangle tests, d-prime was calculated (α-risk = 0.05, β-risk = 0.1, *p*_d_ = 50% for 20 judges).

## 3. Results

### 3.1. Canopy Description

Table 1 shows the differences related to leaf removal. The rows considered had similar characteristics concerning the bunch production (number of bunches on one plant), the shoot density and the percentage of empty spaces. Furthermore, all the shoots completed their active growth before the inspection date, and thus, these data should be considered stable during the whole ripening period. Of course, the leaf removal significantly shortened the leaf wall, mainly due to a significant decrease in primary shoot leaves (and of the related area), with a significant impact on the total leaf area of each shoot and, thus, of the entire plant. It is worth noting that the area of the principal shoot leaves was about 3.5 times bigger than that of secondary shoot leaves: 186.33 ± 31.44 and 53.08 ± 16.00, respectively (*p* = 0.000). Nevertheless, a very limited effect was observed on the secondary shoot leaves; in fact, any parameter related to them was significantly affected by the treatment. Thus, the vegeto-productive balance (leaf area/bunch) was only slightly affected by the treatment.

### 3.2. Micrometeorological Analysis

During the test period (13 July 2021–11 September 2021), the external air temperature varied in the range 13.5–42.0 °C, and the daily average value ranged between 19.8 °C and 31.4 °C, with a mean value of 26.3 °C. The external air relative humidity varied in the range 16.9–100%, and the daily average ranged between 46.2% and 83.1%, with a mean value of 60.3%.

The mean differences in the hourly air temperature (ΔT) and air relative humidity (ΔRH) values in the sun-exposed bunches were tested for significance using t-tests. The results related to the whole day, daytime and night-time datasets are shown in Table 2. ΔT and ΔRH were computed by subtracting the value of the parameter on the row that had not been defoliated from the one on the row that had undergone leaf removal. Daytime hours were considered as those with solar radiation on the horizontal plane greater than zero. Leaf removal caused on average a very slight effect on the air temperature and relative humidity in the bunch, with a slight decrease in the air temperature and an increase in the air relative humidity particularly during night-time. The highest absolute differences were found in the daytime period. Nevertheless, the average air temperatures were very similar among the two bunch exposures. The period was characterized by very poor rainfall: during the whole experimental period, there were only three rainy days with more than 1 mm/24 h and three days with less rain.

The simulation of hourly temperature of exposed berries was performed for the period 13 July–11 September, considering east and west exposition and shaded berries. Figure 2 shows the maximum and minimum daily values of the three considered conditions, together with air measured values. Regarding maximum daily values, west exposition obtains the highest values. Night minimum temperatures are very similar among the different conditions.

In order to summarize the thermal conditions of berries during ripening, the frequency of hourly temperatures during the whole simulation period is represented in Figure 3. The peak frequency is 24 °C for all the conditions, west exposition being the one that reaches the highest classes, and east exposition only slightly lower. In this regard, it is interesting to highlight that the east exposition spent 33.7% of time in the range 30–40 °C against 27% of west exposition, while 7% of time was spent by east at a temperature above 40 °C compared to 8.5% by west exposition. The simulation of the shadowed cluster (50% of shading effect by leaves) is very close to the east one but smoother. All the three conditions differ from air behavior, with a shift toward higher temperatures.

### 3.3. Carpological Analysis and Description of Sunburn Symptoms

Table 3 shows the results of the carpological analysis. It reports the varietal averages, as well as the differences in the fruit characteristics related to leaf removal and light sunburn symptoms. Stronger symptoms were not considered in this elaboration as they were not observed in all the samples.

Leaf removal caused a significant decrease in berry growth (average berry weight), with a consequent increase in the percentage contribution of seeds to the total berry weight.

Despite the similar berry weights of asymptomatic and amber-colored berries, the photo-oxidative sunburn caused a thickening of the skins, with a consequent increase in the percentage contribution of skin to the total berry weight. No difference in the percentage contribution of seeds to the total berry weight was observed, although there was a slightly different number of seeds between the two classes of grapes.

The canopy management influenced the grape quality (Figure 4 and Appendix A). Figure 4a shows the percentage of sunburn damages, calculated based on the number of berries. Of course, due to the lower weight of sunburned berries (especially the severely damaged and the completely dry ones), the contribution of the damaged grapes on the grape weight decreases, but it is still significant (Figure 4b). Finally, Figure 4c shows significant differences among the proportion of solids (skins, seeds and dry grapes) and liquid (pulp), with an expected higher winemaking yield for non-defoliated grapes.

### 3.4. Production of Wines with Non-Sunburned and Sunburned Grapes

Experimental wines were produced with the two batches of grapes with and without maceration. Such oenological approaches could provide evidence for the possible suitability of Verdeca grapes for the production of orange/amber wine as well as the impact of sunburn on the wine composition and characteristics. To the best of our knowledge, this is the first time that both aspects are taken into account for wine production using this grape variety. The potential appropriateness of the Verdeca grape to maceration could exalt this autochthonous grape for differentiating the wine style. With regard to the sunburn, this study would evidence the importance of canopy management to counteract the climatic changes we have been experiencing in the last two decades.

As mentioned above, the strong difference of percentage distribution among skin, pulp and seeds (Section 3.3) affected the must yield, being about 30% lower when pressing sunburned grapes. Nonetheless, the concentrations of sugars were comparable between the two musts (219 ± 1 g/L vs. 209 ± 15 g/L for non-sunburned and sunburned grape musts, respectively). On the contrary, significant differences were found for the content of RAN, resulting lower in the sunburned grape must (111.8 ± 5.6 mg/L) in comparison to the non-sunburned one (158.8 ± 7.9 mg/L). In particular, the ammonium salt content was about 2-fold lower (74.1 ± 3.7 mg/L vs. 34.2 ± 1.7 mg/L for non-sunburned and sunburned grape musts, respectively), while no significant difference was revealed for the amino nitrogen (84.7 ± 4.2 mg/L vs. 77.6 ± 3.9 mg/L for non-sunburned and sunburned grape musts, respectively). In both cases, the content of RAN was adjusted up to 200 mg/L prior to the alcoholic fermentation (AF). Significant differences were found with regard to both the content of flavonoids and the absorbance at 420 nm. In particular, flavonoids were about 30% higher in must obtained from sunburned grapes (201.5 ± 25.3 mg/L of catechin vs. 135.4 ± 9.8 mg/L of catechin for must from non-sunburned grapes). Similarly, a significantly higher color index was observed in must from sunburned grapes (0.71 ± 0.00 AU vs. 0.45 ± 0.12 AU for must from non-sunburned grapes). The trend of AF was monitored for each must (Figure 5). In general, the rates of AF were higher in the winemaking trials with maceration with significant difference between non-sunburned (72.4 ± 0.4 g CO_2_/L produced in tumultuous phase) and sunburned grape musts (67.5 ± 0.9 g CO_2_/L produced in tumultuous phase) (Figure 5B). On the contrary, no differences were observed for the winemaking without maceration (66.4 ± 0.7 g CO_2_/L vs. 63.9 ± 3.9 g CO_2_/L produced in tumultuous phase for non-sunburned and sunburned grape musts, respectively) (Figure 5A). These findings suggest the grape sunburn could not or could only slightly affect the fermentative ability of yeast, with the exception of winemaking with skin maceration.

The multivariate test was significant when the effect of winemaking without/with maceration (*p* = 0.000) and non-sunburned/sunburned grapes (*p* = 0.000) were considered, as well as their combination (*p* = 0.019). Evidence of the limited impact of the grape sunburn on AF was proved considering the residual sugars and the ethanol released (Table 4). In fact, all the fermentations were completed as the residual sugars were lower than 1 g/L with the only exception of the wines produced without maceration with sunburned grapes showing residual sugars of 2.09 ± 1.60 g/L (Table 4). No significant difference was found for the concentrations of ethanol released. The grape sunburn did not influence the titratable acidity, as also supported by the comparable levels of tartaric acid, while the maceration did, as lower values were observed in the wines produced with maceration (Table 4). Surprisingly, higher contents of malic acid and lower concentrations of lactic acid were detected in wine produced with sunburned grapes, with differences also related to the maceration step. Only little differences were found for both acetic and succinic acids; the former acid was significantly higher in the wines produced with maceration of sunburned grape pomace, while the latter was significantly higher in the wines produced with maceration of non-sunburned grape pomace (Table 4). With regard to phenolics, higher concentrations were detected in the macerated wines, as expected. It is interesting to note that the antioxidant capacity values were comparable within wines produced with non-sunburned/sunburned grapes. Nonetheless, relevant differences were revealed when the ratio antioxidant capacity/total phenol index (AC/TPI) were determined with lower values for the sunburned grape-made wines (Table 4). The absorbance at 420 nm, index of yellow color, was significantly higher for wines produced with sunburned grapes for those without maceration (0.18 ± 0.01 AU vs. 0.14 ± 0.01 AU for wines made from non-sunburned grapes) as well as with maceration (0.31 ± 0.03 AU vs. 0.21 ± 0.04 AU for wines made from non-sunburned grapes) (Table 4). These differences in the absorbance values were also significantly perceived through sight (17 correct answers out of 20 judges for wines without maceration; 18 correct answers out of 21 judges for wines with maceration). In particular, a greater difference was estimated for the wines with maceration (d’ = 1.105) in comparison to those without maceration (d’ = 0.907). A significant difference was revealed from the sensory point of view with regard to the wines without maceration: 17 judges out of 20 indicated the correct answer (d’ = 0.907). Such difference was ascribed to a higher perception of acidity in wines made from non-sunburned grapes and a higher perception of oxidized notes for wines made from sunburned grapes. On the contrary, no significant difference was found for the wines produced with maceration (13 correct answers out of 20 judges, d’ = 0.472).

## 4. Discussion

The proper management of the canopy can have a strong impact on the overall characteristics of grapes and, consequently, on the resulting wine. We investigated the effect of leaf removal at veraison on the plant performance, microclimate conditions and wine characteristics. To the best of our knowledge, this expanded approach was applied for the first time and can have a meaningful interest for both the viticulturists and winemakers that are dealing with the extreme climate changes over the past two decades.

The canopy manipulation affected the leaf area in the lower part of the leaf wall, mainly reducing the number of leaves on the principal shoot. Due to the bigger area of these leaves with respect to the ones of the secondary shoots, this result was also reflected in a general decrease in the total leaf area, although the difference in the leaf area of secondary shoots was not significant. Nevertheless, the vegeto-productive equilibrium was kept similar among the two growing conditions (no significant differences were observed concerning the leaf area/bunch). Furthermore, it should be kept in mind that, during ripening, the old basal leaves of the principal shoots are much less photosynthetically efficient than the younger leaves belonging to the secondary shoots [40]. Thus, the differences obtained between the grapes grown in the two canopy conditions should be ascribed mainly to the bunch microclimate.

Regarding the micrometeorological characterization, the direct measurement of air temperature in proximity to the bunches showed similar results for the two canopy managements. Furthermore, it should be noted that the air temperature of the shaded bunches was slightly, but significantly, higher than that of the exposed bunches.

This can be explained considering that shaded bunches are characterized by a smaller sky view factor when compared to exposed ones. This reduces the cooling capability during the night. Furthermore, exposed bunches, due to leaf removal, are more affected by ventilation [41]. The abovementioned effects could overcome the warming of exposed bunches caused by the daily hours of direct insolation, determining the lower inner bunch temperature in the case of the exposed ones.

This evidence suggests that the symptoms of stress recorded in sun-exposed grapes should not be ascribed to the air temperature outside the berries. In fact, despite the similarity in the air temperatures, the irradiance due to the direct sun exposure caused an increase in the temperatures of the berries. The simulation of temperature of berries highlighted similar temperatures for the exposed ones, west exposition being slightly warmer. Obviously, the effect of visible and UV radiation must be added to the thermal effect of irradiance. We can therefore deduce that direct exposure to sun radiation of the berries plays a crucial role in the development of the sunburn symptom. This result is in agreement with those reported by Rustioni et al. [14], in which the central role of radiation in grape sunburn injuries was underlined.

Leaf removal at bunch closure caused a significant increase in sunburn symptomatic berries. It has been shown that grapes are particularly susceptible to photo-oxidative sunburn when exposed to visible light in this phenological phase due to the high concentration of photosynthetic pigments [14]. This shock causes a physiological disorder characterized by dramatic changes in the antioxidant system, in which only a few chemical antioxidant mechanisms could still play a protective role [17]. This results in an accumulation of oxidized polymeric pigments, typical of sunburned tissues, which could play a screening role against the excessive photosynthetically active radiation [12].

Rustioni et al. [16] observed only symptoms of sunburn browning due to photo-oxidative sunburn in an experiment conducted in a vineyard located in Lombardy, North Italy, as a consequence of a short-time sun exposure (5 h) in the last part of the ripening process. The anticipation of the leaf removal in a phenological phase highly reactive to sunburn (bunch closure, with green berries), the experimental site in highly stressful conditions (Mediterranean climate characterized by strong summer stresses for crops) and the long-lasting treatment caused the appearance of both the symptoms of sunburn browning and necrosis as described by Gambetta et al. [13]. In fact, besides amber-colored berries, severely damaged berries were also obtained, which were characterized by dark brown or black-purple necrotic spots on their skins, as well as berries that had undergone cracking and shriveling, which became completely dry. It is worth noting that, in our experimental conditions, all these symptoms (both browning and necrosis) are mainly caused by direct irradiation of the berries and not by environmental conditions external to the grape itself, not even at the microscale level.

Considering the oenological impact of the carpological results, it should be noted that ‘Verdeca’ is characterized by small berries falling in the 20th percentile of the *Vitis vinifera* phenotypic distribution [42]. With respect to this distribution, the berries of ‘Verdeca’ are characterized by a quite low percentage of light skins and a relatively high percentage of big seeds. The canopy management significantly affected the berry size, obtaining smaller berries in sun-exposed grapes. This result suggests that the interaction of each cultivar with the growth conditions, as well as the leaf removal timing, could lead to different responses in the plant. For example, Molitor et al. [43] and Tardaguila et al. [44] did not observe significant modifications in the berry size due to different timing of leaf removal in ‘Grenache’, ‘Sauvignon blanc’, ‘Auxerrois’, ‘Pinot gris’ and ‘Riesling’, while Piombino et al. [20] reported an increase in the ‘Nebbiolo’ berry mass when bunches were sun-exposed at the fruit-set. Sunburn caused a thickening of the skins of amber-colored berries, probably due to the protective role of the exocarp against stresses. Thus, the increased number of completely dry berries and the contribution of amber berries (with thicker skins) caused, in sun-exposed grapes, an increase in the proportion of solid parts with respect to the liquid ones. This disproportion between solid and liquid parts can obviously have a noteworthy impact on the wine production as it is directly correlated to the winemaking yield, a main objective of wine producers since the birth of oenology [45]. Indeed, a lower must extraction yield was found in the case of sun-exposed grapes.

With regard to the grape composition, the major light exposure caused a decrease of ammonium salt, and consequently of RAN, that could be due to the higher temperature reached when leaf removal was carried out. Friedel et al. [46] observed a decrease of nitrogen in ‘Riesling’ grapes related to higher temperature. Nonetheless, the compounds of oenological interest (e.g., phenolics) are highly concentrated in solid part and their concentrations can further increase as a consequence of considerable exposure to sunlight. In fact, considering the flavonoids determined after grape pressing, higher content was found in must from sunburned grapes, leading to a higher color index as well. Both these parameters were about 30% higher in comparison to the must from non-sunburned grapes, possibly dependent on the lower must yield extraction as well. This result could suggest easier extraction of phenolics in sunburned grapes, even if further study should be carried out in order to confirm this hypothesis. For both the vinification approaches (without/with maceration) adopted in this study, the content of phenols was higher in the sunburned grape-made wines, but an unbalanced ratio AC/TPI towards phenols was revealed when sun-exposed grapes were used for the wine production without maceration. This indicates the oxidative impact played by the sunburn on phenolics and, possibly, on other antioxidant compounds (e.g., glutathione, ascorbic acid), as previously reported for grape skin by Rustioni et al. [17]. The general chemical parameters of wine (i.e., pH, titrable acidity, ethanol) were not affected by the use of sunburned grapes. Significant differences were detected, even within the trials with and without maceration, for both malic acid and lactic acid, possibly due to the occurrence of malolactic fermentation. We did not monitor the microbiome of must before and during the fermentation; consequently, we cannot exclude the possible growth of lactic acid bacteria (LAB). Nonetheless, the grape sunburn seemed to limit the development of LAB since lower concentrations of lactic acid were found in wines produced with sunburned grapes (both with and without maceration). Further investigations will be carried out to confirm this hypothesis. Even if lower content of malic acid was found in wines produced with non-sunburned grapes, the main sensory attribute indicated as different for wines produced without maceration was the perception of acidity. This cannot be explained by the titratable acidity or pH as they were not significantly different. Moreover, the d’ slightly lower than one (0.907) indicated the perceived difference among the wines obtained with non-sunburned and sunburned grapes. In these wines, the higher ratio AC/TPI as well as the lower absorbance value at 420 nm can be related to a lower oxidative degree of wine phenolics in comparison to the wines obtained from sunburned grapes. The major intensity of the yellow color indicates greater oxidation that can be associated with the complex sunburn-dependent oxidative phenomena [14,17]. On the contrary, no significant difference was revealed for wines produced with maceration when the taste was considered with d’ < 1 (0.472). The d’ value data suggests there is limited difference in taste between the two wines produced with maceration, while relevant difference in wine color was observed as d’ > 1 (1.105). On the contrary, no significant difference was revealed for wines produced with maceration. Maybe the higher content of phenolics could affect the sensory characteristics of wines since phenols can suppress, accentuate or show negligible effect on the perception of the aroma compounds [47]. The higher phenolic concentration not associated with a higher astringency perception observed in wines produced by maceration of sunburned grapes could be due to the oxidative polymerizations of tannins. This kind of oxidative reaction typically occurs during both ripening and sunburn [17], and it has already been proposed as a technique able to enhance the phenolic ripening in grape seeds in difficult growing conditions [48].

## 5. Conclusions

Nowadays, wineries should deal with the important impacts of climate change on the production chain. A systemic approach is necessary to optimize the interconnections between crop growing and food processing. Different microclimatic growing conditions affect the crop production. Summer stresses, including excesses of berry irradiation, significantly affect grape quality, often leading sunburn symptoms. This could result in the accumulation of oxidized polymeric pigments (sunburn browning) or in tissue necrosis, also associated with berry cracking and desiccation. The modified quality of grapes affects the winemaking processing, which should be set up aiming to maximize the wine quality, taking into consideration the grape quality and the different processing options.

This multidisciplinary work highlights the importance of a systemic approach to the decision-making process of the production chain, testing the interactions among vineyard management (bunch exposure to direct sunlight) and winemaking decisions (maceration).

This study highlights the impact of solar irradiance on grape carpological characteristics and quality. This can influence the content of phenolics and the antioxidant capacity in interaction with the winemaking decisions and lead to differences in acidity, pH and wine color, thus significantly affecting the quality of the final product. In details, it is worth noting that the canopy management significantly affected the sensory characteristics of wine when it was obtained using white winemaking (without maceration), showing a lower perception of acidity and a higher oxidized character. On the contrary, when the wines were made with maceration (orange/amber wines), no significant differences were perceived in terms of taste-tactile and flavor perception. In this case, the chemical analyses showed significantly higher titratable acidity and phenolic content, not associated with a higher astringency perception. This could be due to the phenolics oxidative polymerization. Thus, it seems that this vinification style could be adapted to make wines from grapes that have undergone oxidative stress in the vineyard, not showing significant worsening of the wine quality due to sunburn symptoms. Further studies will be carried out considering other white grape varieties in order to confirm this hypothesis.

## Figures and Tables

**Figure 1 foods-12-00621-f001:**
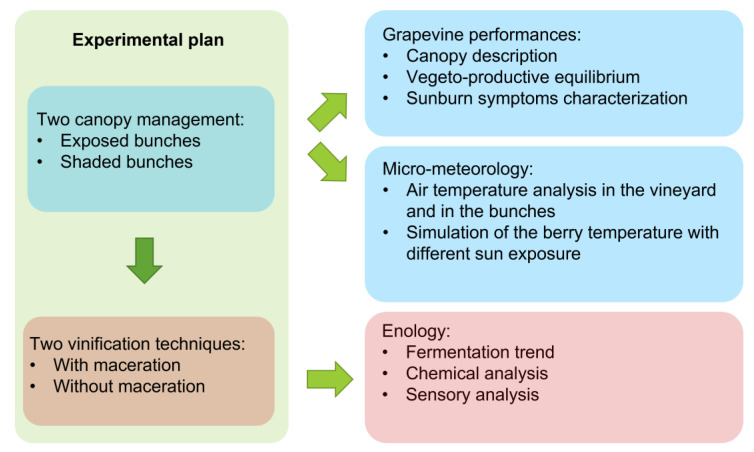
Flow chart of the experimental plan.

**Figure 2 foods-12-00621-f002:**
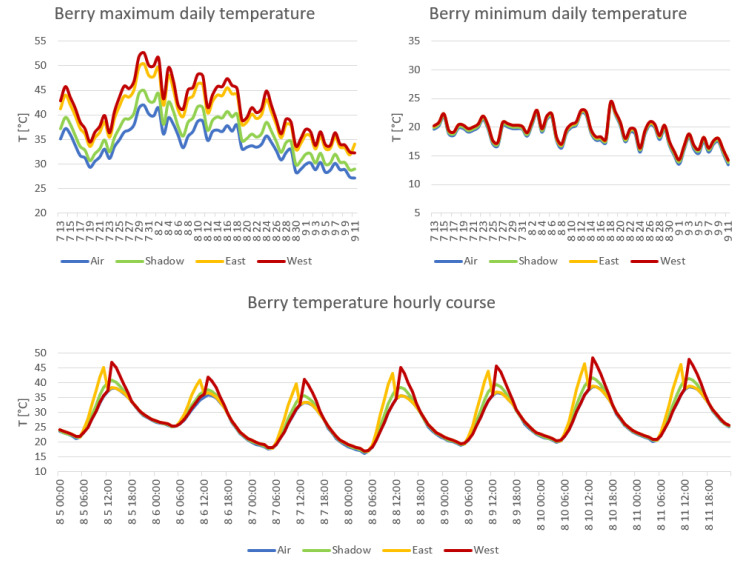
Simulation of exposed berry temperature.

**Figure 3 foods-12-00621-f003:**
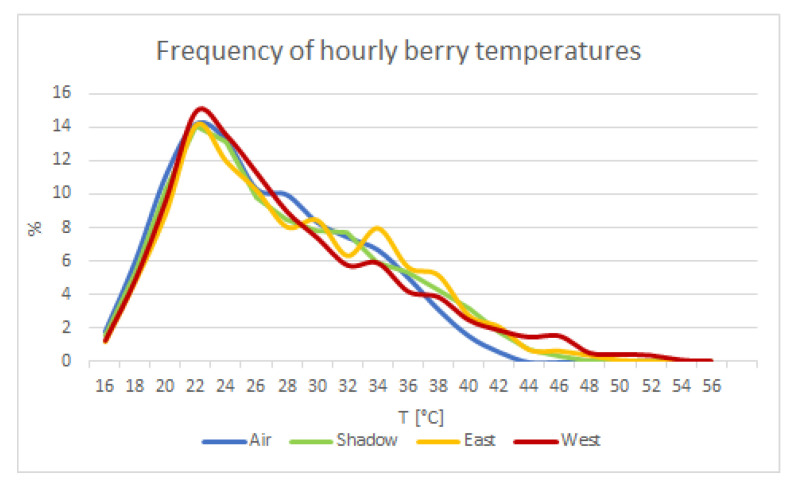
Distribution of exposed berry temperature along the period 13 July–11 September. A focus on hourly behavior was provided for the period 5–11 August.

**Figure 4 foods-12-00621-f004:**
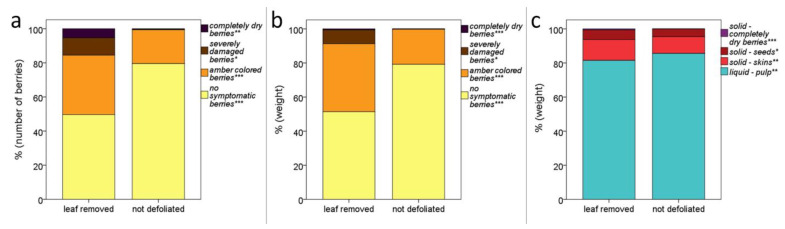
Effects of the canopy management on grape quality. (**a**) Proportion among the number of berries affected by different intensity of sunburn symptoms. (**b**) Proportion among the weight of grapes affected by different intensity of sunburn symptoms. (**c**) Expected ratio between liquid and solids during winemaking. * *p* < 0.05; ** *p* < 0.01; *** *p* ≤ 0.005.

**Figure 5 foods-12-00621-f005:**
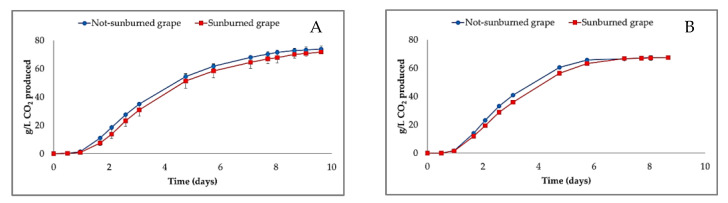
Fermentation trend of wine produced (**A**) without and (**B**) with maceration. Error bars indicate the variation observed (n = 3).

**Table 1 foods-12-00621-t001:** Results of the canopy description and of the vegeto-productive balance.

Parameter	Leaf Removed	Not Defoliated	Significance of the Difference *
Leaf wall height (cm)	**87.13** ± 10.73	**111.20** ± 13.26	**0.003**
Percentage of empty spaces (%)	23.13 ± 14.00	20.80 ± 15.47	0.336
Number of leaf layers	3.36 ± 1.23	3.84 ± 1.28	0.162
Shoot density (number/m of row)	16.05 ± 4.25	13.92 ± 2.59	0.179
Number of bunches in one plant	15.93 ± 5.96	16.07 ± 6.68	0.818
Number of shoots in active growth	0.000	0.000	
Number of primary shoots in one plant	12.87 ± 3.29	12.87 ± 3.76	0.628
Number of secondary shoots in one primary shoot	6.73 ± 3.22	9.00 ± 2.10	0.144
Number of primary shoot leaves in one shoot	**7.73** ± 1.87	**12.87** ± 2.45	**0.002**
Number of secondary shoot leaves in one shoot	21.27 ± 17.54	27.33 ± 18.02	0.314
Area of primary shoot leaves in one shoot (cm^2^)	**1440.98** ± 348.36	**2397.49** ± 455.70	**0.002**
Area of secondary shoot leaves in one shoot (cm^2^)	1128.84 ± 931.22	1450.85 ± 956.21	0.314
Total leaf area of one shoot (cm^2^)	**2569.81** ± 946.24	**3848.34** ± 855.31	**0.001**
Total leaf area of one plant (m^2^)	**3.20** ± 1.03	**4.86** ± 1.39	**0.021**
Percentage of leaf area due to primary shoot leaves (%)	61.04 ± 19.14	64.30 ± 15.35	0.784
Percentage of leaf area due to secondary shoot leaves (%)	38.96 ± 19.14	35.71 ± 15.35	0.784
Leaf area/bunch (m^2^)	0.24 ± 0.17	0.35 ± 0.17	0.065

Results are average ± standard deviation (n = 5). * *p* values; significant mean differences are written in bold.

**Table 2 foods-12-00621-t002:** Maximum, minimum and mean hourly differences between air temperature (ΔT) and air relative humidity (ΔRH) values in the bunch due to leaf removal.

		ΔT	ΔRH
Whole day	Mean	**−0.6**	**1.3**
	S.E. mean	0.0	0.1
	Sign. (*p* value)	**0.000**	**0.000**
	Max.	3.0	20.3
	Min.	−5.0	−15.3
Daytime	Mean	**−0.5**	**0.1**
	S.E. mean	0.0	0.1
	Sign. (*p* value)	**0.000**	0.313
	Max.	3.0	20.3
	Min.	−5.0	−15.3
Night-time	Mean	**−0.7**	**3.2**
	S.E. mean	0.0	0.1
	Sign. (*p* value)	**0.000**	**0.000**
	Max.	0.5	13.6
	Min.	−3.0	−7.4

Significant mean differences are written in bold. Legend: max., maximum; min., minimum; S.E. mean, standard error of the mean; sign., *t*-test significance.

**Table 3 foods-12-00621-t003:** Results of the carpological analysis and of sunburn symptoms.

Parameter	Varietal Average ^1^	Leaf Removal Effect			Light Sunburn Symptom Effect		
		Leaf Removed	Not Defoliated	Significance ^2^	Asymptomatic Berries	Amber-Colored Berries	Significance ^2^
Average berry weight (g)	1.49 ± 0.21	**1.32** ± 0.15	**1.66** ± 0.08	**0.001**	1.45 ± 0.26	1.53 ± 0.17	0.223
Skin weight (g)	0.16 ± 0.03	0.15 ± 0.03	0.17 ± 0.02	0.126	**0.15** ± 0.01	**0.18** ± 0.03	**0.031**
Number of seeds per berry	1.82 ± 0.25	1.82 ± 0.35	1.82 ± 0.12	1.000	**1.67** ± 0.23	**1.97** ± 0.16	**0.039**
Seed weight (mg)	41.61 ± 2.80	40.79 ± 3.09	42.43 ± 2.47	0.357	41.52 ± 2.87	41.69 ± 3.00	0.920
% of skin	11.12 ± 1.33	11.66 ± 0.98	10.57 ± 1.50	0.084	**10.47** ± 1.47	**11.76** ± 0.88	**0.048**
% of seeds	5.11 ± 0.72	**5.58** ± 0.69	**4.65** ± 0.39	**0.011**	4.83 ± 0.69	5.40 ± 0.69	0.076

Results are average ± standard deviation (n = 5). ^1^ Considering only asymptomatic and amber-colored berries. ^2^ *p* values; significant mean differences are written in bold.

**Table 4 foods-12-00621-t004:** Chemical composition of wines produced without maceration and with maceration from non-sunburned and sunburned grapes.

Parameter	Vinification Without Maceration		Vinification with Maceration	
	Non-Sunburned Grapes	Sunburned Grapes	Non-Sunburned Grapes	Sunburned Grapes
Residual sugars (g/L)	0.37 ± 0.01 aA	2.09 ± 1.60 bA	0.77 ± 0.24 aA	0.46 ± 0.10 aB
pH	3.34 ± 0.01 aA	3.40 ± 0.11 aA	3.85 ± 0.25 aB	3.58 ± 0.01 bA
Titratable acidity (g tartaric acid/L)	5.3 ± 0.4 aA	5.3 ± 0.6 aA	4.3 ± 0.1 aB	4.6 ± 0.1 aB
Ethanol (%, *v*/*v*)	10.9 ± 0.4 aA	11.0 ± 0.1 aA	9.9 ± 1.2 aA	10.1 ± 0.2 aA
Total phenol index (g gallic acid/L)	265 ± 17 aA	321 ± 12 aA	497 ± 75 aB	707 ± 29 bB
Total flavonoids (g catechin/L)	nm	nm	138 ± 27 a	285 ± 17 b
ABS 420 nm (AU)	0.14 ± 0.01 aA	0.18 ± 0.01 bA	0.21 ± 0.04 aB	0.31 ± 0.03 bB
Antioxidant capacity (mmol Trolox/L)	0.88 ± 0.07 aA	0.89 ± 0.07 aA	2.28 ± 0.036 aB	3.11 ± 0.62 bB
Ratio AC/TPI	3.32 aA	2.78 bA	4.60 aB	4.41 aB
Tartaric acid (g/L)	3.42 ± 0.38 aA	2.84 ± 0.67 aA	1.77 ± 0.03 aB	1.70 ± 0.16 aB
Malic acid (g/L)	1.64 ± 0.26 aA	2.00 ± 0.15 bA	1.71 ± 0.14 aA	2.55 ± 0.11 bB
Lactic acid (g/L)	0.89 ± 0.00 aA	0.44 ± 0.07 bA	1.52 ± 0.08 aB	0.83 ± 0.08 bB
Acetic acid (g/L)	0.47 ± 0.01 aA	0.47 ± 0.05 aA	0.34 ± 0.04 aB	0.66 ± 0.08 bB
Citric acid (g/L)	nd	nd	nd	nd
Succinic acid (g/L)	1.55 ± 0.01 aA	1.64 ± 0.06 aA	1.79 ± 0.05 aB	1.73 ± 0.08 aA

Data are expressed as average ± standard deviation (n = 3). Different lowercase letters mean significant differences related to non-sunburned/sunburned grapes, applying the same winemaking procedure (F test, α = 0.05). Different capital letters mean significant differences related to without/with maceration for the same grape batch (F test, α = 0.05). Legend: nd, not detected; nm, not measured.

## Data Availability

Corresponding authors will provide data if necessary.

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
