# Peer review of "Microclimate of Grape Bunch and Sunburn of White Grape Berries: Effect on Wine Quality"

_foods, 2023, doi:10.3390/foods12030621_

Round 1

Reviewer 1 Report

Manuscripts reads well, but some concerns below

L43 Rather than bullet points, summarise this into a sentence.

L112 It's unclear to me what this information is bringing to the readers?

Due to the complexity of measures, i'd recommend the author to create a flow chart to summarise their experimental plan and measurements for each of it?

Considering that this is a biological sample, how did the author control such variation? Was there any power analysis carried out here?

For triangle test, consider calculating d-prime as well as an estimate on how different the samples are.

There's a posthoc comparison here being carried out, this needs to be added to the stats analysis section.

Figure 4. ANCOVA can be suitable here to observe if there is actually a significant difference 

There's 2 factors being considered, which is maceration, and sunburnt. So isn't 2 way ANOVA more appropriate?

Author Response

Legend: RE, reviewer; AU, authors

RE: Manuscripts reads well, but some concerns below

AU: We thank the referee for the time spent in critically review our manuscript and we did our best to take into consideration all the suggestions received.

RE: L43 Rather than bullet points, summarise this into a sentence.

AU: The sentence has been rewritten following the referee suggestion (lines 47-57): “Generally, during sunburn, photosynthetic pigments undergo degradation [14]. Light and, in general, stresses induce the accumulation of secondary metabolites, such as phenolics [15] and the oxidative stress associated with a cell decompartmentalization cause phenolic oxidation. This lead to the accumulation of brown pigments [12,16]. Thus, changes in the metabolisms induce differences in the chemical composition of the grapes, with impacts on the quality of production [17]. In sunburned grapes, the crystalline structure of the waxes undergo degradation to amorphous masses affecting the protective role of this layer [18]. As a consequence, the loss in wax protection properties and the tissues necrosis (with consequent cracking), favor berry desiccation [13]. In extreme cases, shriveling of entire berries, and even of entire bunches, can occur, also affecting parts of the rachis [13].”

RE: L112 It's unclear to me what this information is bringing to the readers?

AU: In field experiments, it is mandatory to work with a sufficient quantity of plant materials to ensure a correct evaluation of the biological variability. The experiment set-up should be clearly reported, also to demonstrate a correct consideration of the environmental variability within the same vineyard. The timing of the activities is necessary to clarify the management in different phenological phases and, thus, with different physiological impacts. For these reasons we trust that this information should be reported in the text.

RE: Due to the complexity of measures, i'd recommend the author to create a flow chart to summarise their experimental plan and measurements for each of it?

AU: A flow chart has been included (Figure 1).

RE: Considering that this is a biological sample, how did the author control such variation? Was there any power analysis carried out here?

AU: The experimental design is clearly reported in the materials and methods section. The length/distribution of the rows with different canopy management were set-up to ensure a sufficient quantity of plant materials for a correct evaluation of the biological (lines 116). The number of plants studied for the different analyses are also reported (lines 131-146; 164-177), as well as the grape sampling for winemaking (lines 179-182). If a specific point is not enough clear, please, let us know your doubt and we will be glad to improve the text.

RE: For triangle test, consider calculating d-prime as well as an estimate on how different the samples are.

AU: As the reviewer suggested, d-prime was calculated for the triangle tests considering α = 0.05, β = 0.1, pd = 50% (line 271). Based on the d’ values, some considerations regarding the differences between samples were added in the manuscript as reported as follows: “In particular, a greater difference was estimated for the wines with maceration (d’ = 1.105) in comparison to those without maceration (d’ = 0.907).” (lines 425-426). “Moreover, the d’ slightly lower than 1 (0.907) indicated the perceived difference among the wines obtained with not sunburn and sunburn grapes.” (lines 547-548). “On the contrary, no significant difference was revealed for wines produced with macera-tion when the taste was considered with d’ < 1 (0.472). The d’ value data suggests the limited difference of taste between the two wines produced with maceration, while relevant difference in wine color was observed as d’ > 1 (1.105).” (lines 552-556).

RE: There's a posthoc comparison here being carried out, this needs to be added to the stats analysis section.

AU: As the reviewer suggested, more details were reported the Paragraph 2.8 (Statistical analysis) as mentioned as follows: The statistical analysis was performed with SPSS Win 12.0 program (SPSS Inc., Chicago, IL, USA). One-way ANOVA was carried out for the assessment of significant differences related to the grape berries. Factorial ANOVA was carried out to determine the significant differences related to fermentation trend and the chemical parameters of experimental wines. Firstly, the multivariate test of significance was determined; secondly, the post-hoc Fischer LSD (α = 0.05) was carried out. For the triangle tests, d-prime was calculated (α-risk = 0.05, β-risk = 0.1, pd = 50% for 20 judges). (lines 265-271).

RE: Figure 4. ANOVA can be suitable here to observe if there is actually a significant difference 

AU: We are grateful with the reviewer for this comment. The statistical analysis was carried out, but this was not mentioned in the manuscript. We specified this aspect in Paragraph 2.8 (Statistical analysis), lines 265-271, as mentioned in the previous comment. Moreover, we implement this aspect in Results section as well, where the significant difference between the alcoholic fermentation trends in the winemaking with maceration was specified (line 388).

RE: There's 2 factors being considered, which is maceration, and sunburnt. So isn't 2 way ANOVA more appropriate?

AU: We only mentioned the one-way ANOVA in the text. Nonetheless, for the data related to the experimental wine, the factorial ANOVA was considered. This allowed to separately understand the significant differences related to the sunburn and the maceration. 

Reviewer 2 Report

The manuscript is interesting and showed good goals to understand the quality of grape berries and the wine.

In Abstract, the author can detail the experimental procedures used in this project (Material and Methods section), reducing aspects of literature review and introduction subjects

Line 305-306: review this sentence

In Conclusion, the authors can improve and explain the main goals obtained by this study, in a more specific way, not as general as it was presented.

Author Response

Legend: RE, reviewer; AU, authors

RE: The manuscript is interesting and showed good goals to understand the quality of grape berries and the wine.

AU: We thank the referee for the time spent in critically review our manuscript and we did our best to take into consideration all the suggestions received.

RE: In Abstract, the author can detail the experimental procedures used in this project (Material and Methods section), reducing aspects of literature review and introduction subjects

AU: The abstract has been revised, adding the main experimental information (lines 17-20): “In vintage 2021, the canopy of ‘Verdeca’ grapevines grown in Salento (South Italy) was differently managed by sun exposing or shading the bunches. Micrometeorological conditions were studied at different levels. Grapes were vinified comparing the winemaking without and with skin maceration.”

RE: Line 305-306: review this sentence 

AU: The sentence has been revised (lines 321-322) as follows: ““In order to summarize the thermal conditions of berries during ripening, the frequency of hourly temperatures during the whole simulation period is represented in Figure 3.”

RE: In Conclusion, the authors can improve and explain the main goals obtained by this study, in a more specific way, not as general as it was presented.

AU: The conclusions have been revised, following the referee instructions (lines 582-593): “In details, it is worth to notice that the canopy management significantly affected the sensory characteristics of wine when it was obtained by the white winemaking (with-out maceration), showing a lower perception of acidity and a higher oxidized character. On the contrary, when the wines were made with maceration (orange/amber wines), no significant differences were perceived in terms of taste-tactile and flavor perception. In this case, the chemical analyses showed significantly higher titratable acidity and phenolic content, not associated to a higher astringency perception. This could be due to the phenolics oxidative polymerization. Thus, it seems that this vinification style could be adapted to make wines from grapes undergone oxidative stress in the vineyard, not showing significant worsening of the wine quality due to sunburn symptoms. Further studies will be carried out considering other white grape varieties in order to confirm such hypothesis.”

Reviewer 3 Report

Manuscript ID: foods-2166614

The manuscript entitled “Microclimate of grape bunch and sunburn of grape berries: effect on wine quality” concerns the effect study of white wines composition obtained with grapes showing sunburn symptoms as well as showing a complex physiological dysfunction leading to browning or necrosis of berry tissues. The main aim of presented investigations was to evidence the interactions between viticulture and oenological decisions. Two canopy managements performed in an extremely stressing growing area, inducing bunch exposure and bunch shading respectively and two winemaking techniques with and without skin and seed maceration were compared. The authors drew attention to the impact of notable sunlight exposure was evaluated in grape bunches in relation to the microclimate.

I recommend the minor revision according to specific comments:

·   the abstract section lacks important details of obtained results and conclusions from the conducted research, more data should be presented. Particular quantitative results should be taken into account. It should be made clear in this section that mentioned investigations concerns white wines samples, which are susceptible to oxidation processes.

·         Chapter Results and discussion - the critical evaluation of more important quantitative merits (such as LOD, LOQ, sensitivity, linear range, accuracy, etc) is required. There is no information whether the proposed methods have been validated. Presented results of raw plant material quantitative analysis are also missing. It should be completed in the manuscript.

·         conclusions do not indicate elements of scientific novelty. The most important achievements and future perspectives should be indicated.

Summary:

The current version of the manuscript still needs minor revision.

Author Response

Legend: RE, reviewer; AU, authors

RE: The manuscript entitled “Microclimate of grape bunch and sunburn of grape berries: effect on wine quality” concerns the effect study of white wines composition obtained with grapes showing sunburn symptoms as well as showing a complex physiological dysfunction leading to browning or necrosis of berry tissues. The main aim of presented investigations was to evidence the interactions between viticulture and oenological decisions. Two canopy managements performed in an extremely stressing growing area, inducing bunch exposure and bunch shading respectively and two winemaking techniques with and without skin and seed maceration were compared. The authors drew attention to the impact of notable sunlight exposure was evaluated in grape bunches in relation to the microclimate.

AU: We thank the referee for the time spent in critically review our manuscript and we did our best to take into consideration all the suggestions received.

RE: I recommend the minor revision according to specific comments:

the abstract section lacks important details of obtained results and conclusions from the conducted research, more data should be presented. Particular quantitative results should be taken into account. It should be made clear in this section that mentioned investigations concerns white wines samples, which are susceptible to oxidation processes.

AU: The abstract has been revised implementing the experiment description and the result section. However, due to the complexity of the experimental plan, it is not easy to include quantitative numbers in the summary. Moreover, we added the adjective ‘white’ in the manuscript title.

RE: Chapter Results and discussion - the critical evaluation of more important quantitative merits (such as LOD, LOQ, sensitivity, linear range, accuracy, etc) is required. There is no information whether the proposed methods have been validated. Presented results of raw plant material quantitative analysis are also missing. It should be completed in the manuscript.

AU: As the reviewer suggested, more details regarding the applied methods were reported. In particular, the accuracy, expressed as relative standard deviation (RDS%), of the enzymatic kits is equal to 1.3% for d-fructose/d-glucose, ammonium kits, and 1.5% for α-amino nitrogen (lines 210-212). For the chromatographic method used for the determination of organic acids, a linear response was obtained in the concentration range 0.1-10 g/L. The detection limit (LOD) and the quantification limit (LOQ) of the method were 0.003 g/L and 0.010 g/L, respectively for LOD and LOQ. RSD% ranged from 3.4% for tartaric acid to 8.6% for succinic acid.” (lines 248-251).

The objective of the current study was to clarify the impact of wine composition when sunburn grape is used for the wine production. For this reason, the chemical characterization of grape berries was not carried out. Nonetheless, the composition of must obtained after the pressing of both not sunburn and sunburn grapes was taken into account. In particular, the parameters being of important for the fermentation were assessed, including sugars, ammonium salt and amino nitrogen, the latter two allowing the assessment of the readily assimilable nitrogen (lines 373-380). Moreover, the manuscript was implemented as the data related to the total flavonoids and the color index were also reported in the manuscript in lines 382-386 as reported as follows: “Significant differences were found with regard to both the content of flavonoids and the absorbance at 420 nm. In particular, flavonoids were about 30% higher in must obtained from sunburn grape (201.5±25.3 mg/L of catechin vs. 135.4±9.8 mg/L of catechin for must from not-sunburn grape). Similarly, significantly higher color index was observed in must from sunburn grape (0.71±0.00 AU vs. 0.45±0.12 AU for must from not-sunburn grape).” This data was discussed in lines 523-529: “In fact, considering the flavonoids determined after grape pressing, higher content was found in must from sunburn grape leading to a higher color index as well. Both these pa-rameters were about 30% higher in comparison to the must from not sunburn grape may-be dependent to the lower must yield extraction, as well. This result could suggest the eas-ier extraction of phenolics in sunburn grape, even if further study should be carried out in order to confirm such hypothesis.”

RE: conclusions do not indicate elements of scientific novelty. The most important achievements and future perspectives should be indicated.

AU: The conclusions have been revised, following the referee instructions (lines 582-593).

“In details, it is worth to notice that the canopy management significantly affected the sensory characteristics of wine when it was obtained by the white winemaking (with-out maceration), showing a lower perception of acidity and a higher oxidized character. On the contrary, when the wines were made with maceration (orange/amber wines), no significant differences were perceived in terms of taste-tactile and flavor perception. In this case, the chemical analyses showed significantly higher titratable acidity and phenolic content, not associated to a higher astringency perception. This could be due to the phenolics oxidative polymerization. Thus, it seems that this vinification style could be adapted to make wines from grapes undergone oxidative stress in the vineyard, not showing significant worsening of the wine quality due to sunburn symptoms. Further studies will be carried out considering other white grape varieties in order to confirm such hypothesis.”